# Antibacterial Films of Silver Nanoparticles Embedded into Carboxymethylcellulose/Chitosan Multilayers on Nanoporous Silicon: A Layer-by-Layer Assembly Approach Comparing Dip and Spin Coating

**DOI:** 10.3390/ijms241310595

**Published:** 2023-06-24

**Authors:** Nelson Naveas, Ruth Pulido, Vicente Torres-Costa, Fernando Agulló-Rueda, Mauricio Santibáñez, Francisco Malano, Gonzalo Recio-Sánchez, Karla A. Garrido-Miranda, Miguel Manso-Silván, Jacobo Hernández-Montelongo

**Affiliations:** 1Departamento de Física Aplicada, Universidad Autónoma de Madrid, 28049 Madrid, Spain; ruth.pulido@estudiante.uam.es (R.P.); vicente.torres@uam.es (V.T.-C.); miguel.manso@uam.es (M.M.-S.); 2Departamento de Ingeniería Química y Procesos de Minerales, Universidad de Antofagasta, Antofagasta 1270300, Chile; 3Departamento de Química, Universidad de Antofagasta, Avda. Universidad de Antofagasta 02800, Antofagasta 1240000, Chile; 4Instituto de Ciencia de Materiales de Madrid (ICMM), Consejo Superior de Investigaciones Científicas (CSIC), 28049 Madrid, Spain; far@icmm.csic.es; 5Departamento de Ciencias Físicas, Universidad de la Frontera, Temuco 4811230, Chile; mauricio.santibanez@ufrontera.cl (M.S.); francisco.malano@ufrontera.cl (F.M.); 6Centro de Excelencia en Física e Ingeniería en Salud (CFIS), Universidad de La Frontera, Temuco 4811230, Chile; 7Facultad de Ingeniería, Arquitectura y Diseño, Universidad de San Sebastián, Concepción 4080871, Chile; gonzalo.recio@uss.cl; 8Núcleo Científico y Tecnológico de Biorecursos (BIOREN), Universidad de La Frontera, Temuco 4811230, Chile; karla.garrido@ufrontera.cl; 9Centro de Genómica Nutricional Agroacuícola (CGNA), Temuco 4780000, Chile; 10Departamento de Ciencias Matemáticas y Físicas, Universidad Católica de Temuco, Temuco 4813302, Chile; 11Departamento de Bioingeniería Traslacional, Universidad de Guadalajara, Guadalajara 44430, Mexico

**Keywords:** antibacterial films, carboxymethylcellulose, chitosan, silver nanoparticles, nanoporous silicon, composite material, layer-by-layer

## Abstract

The design and engineering of antibacterial materials are key for preventing bacterial adherence and proliferation in biomedical and household instruments. Silver nanoparticles (AgNPs) and chitosan (CHI) are broad-spectrum antibacterial materials with different properties whose combined application is currently under optimization. This study proposes the formation of antibacterial films with AgNPs embedded in carboxymethylcellulose/chitosan multilayers by the layer-by-layer (LbL) method. The films were deposited onto nanoporous silicon (nPSi), an ideal platform for bioengineering applications due to its biocompatibility, biodegradability, and bioresorbability. We focused on two alternative multilayer deposition processes: cyclic dip coating (CDC) and cyclic spin coating (CSC). The physicochemical properties of the films were the subject of microscopic, microstructural, and surface–interface analyses. The antibacterial activity of each film was investigated against *Escherichia coli* (Gram-negative) and *Staphylococcus aureus* (Gram-positive) bacteria strains as model microorganisms. According to the findings, the CDC technique produced multilayer films with higher antibacterial activity for both bacteria compared to the CSC method. Bacteria adhesion inhibition was observed from only three cycles. The developed AgNPs–multilayer composite film offers advantageous antibacterial properties for biomedical applications.

## 1. Introduction

According to the International Union of Pure and Applied Chemistry (IUPAC), a biofilm is an aggregate of microorganisms in which the forming units adhere to each other and/or to a surface. This is frequently achieved by embedding within a self-produced matrix of extracellular polymeric substances (EPSs) [1]. The bacteria in biofilms are significantly more tolerant to antibiotics, biocides, and environmental stress, such as the immune system attack of a host organism. Furthermore, the proximity between units in biofilm facilitates the exchange of gene transfer and metabolized products [2]. Therefore, to prevent the proliferation of microorganisms in different hosts, it is important to avoid adhesion to surfaces. This strategy includes reducing surface roughness or anti-adhesive compounds that repel bacterial cells using physical mechanisms [3]. However, providing antimicrobial activity on surfaces is desirable to prevent clinical infections in severe surgical practice. This protection can be proportionated by antibiotics, antimicrobial proteins, enzymes, quorum-sensing inhibitors, nanoparticles, and other methods [4]. In this regard, one of the most versatile and interesting techniques for obtaining antibacterial surfaces for biotechnological applications is the layer-by-layer technique (LbL) [5]. LbL techniques alternate the physisorption of oppositely charged polyelectrolytes, whereby electrostatic interactions between molecules are critical to controlling stability/degradability [6]. The main aim of the LbL technique is the multilayer assembly of polymers, colloids, or biomolecules on substrates to form a surface coating that provides superior control and versatility to other film deposition methods, especially on micro- and nanostructured surfaces [7]. In this sense, nanoporous silicon (nPSi) can be an excellent biomaterial for LbL assemblies due to its biocompatibility, biodegradability, and bioresorbability [8]. nPSi consists of silicon nanocrystals embedded in a porous amorphous matrix. It has been successfully used in many in vitro and in vivo applications, such as drug delivery, diagnosis, imaging, complex biochip systems, and tissue engineering [9].

On the other hand, various synthetic polymers can be used to prepare multilayers but often lack bioactivity and biocompatibility, resulting in adverse side effects [10]. In this context, due to its natural origin, biocompatibility, biodegradability, and non-cytotoxicity, chitosan (CHI) is an excellent antibacterial candidate [11]. The antibacterial action of CHI is mainly due to the electrostatic interaction between its protonated amino groups, positively charged in acidic and moderately acidic media, and the negatively charged bacterial membrane, resulting in the leakage of the cytoplasmic content and, consequently, the death of the microorganisms [12]. However, CHI is less effective against Gram-negative bacteria than Gram-positive bacteria because they are protected by the outer cell membrane that constitutes the outer surface of the cell wall and can hinder its interaction with CHI [13]. In this instance, the addition of active antimicrobial species, for example, antimicrobial peptides or silver nanoparticles (AgNPs), is recommended. AgNPs are well-known antibacterial agents with low toxicity towards mammalian cells at controlled concentrations [14,15]. The antibacterial mechanism of AgNPs is related to their easy adsorption to the membrane surface of bacteria through electrostatic interactions, forming permeable pits and causing an osmotic collapse in the cells [15]. CHI and Ag can be combined by CHI’s sorption abilities and capacity to form chelate compounds with metal ions, such as Ag^+^ [16], which could generate synergistic antibacterial effects. Thus, CHI-Ag, a polycation complex, can be assembled LbL with carboxymethylcellulose (CMC), a water-soluble cellulose-derived polyanion [17].

The five most used routes of assembling films LbL are: (i) immersive or dip, (ii) spin, (iii) spray, (iv) electromagnetic, and (v) fluidic assembly, with dip-coating (D.C.) and spin-coating (S.C.) being the most frequently employed because they are simple, versatile, and cost-effective for the fabrication of thin-film coatings [6,18].

Based on the aforementioned considerations, the primary objective of this study is to synthesize antibacterial films incorporating AgNPs embedded within CMC/CHI multilayers assembled on nPSi. The fabrication process involved two distinct LbL routes, namely dip-coating and spin-coating. Comprehensive characterization of the resulting samples obtained from both techniques was conducted, focusing on their structural and physicochemical properties. Subsequently, the antibacterial efficacy of these films was systematically evaluated against *Escherichia coli* (Gram-negative) and *Staphylococcus aureus* (Gram-positive) bacteria strains, which served as representative microorganisms in this study. By providing insights into the antimicrobial activity of these films, this research contributes to the development of effective antibacterial coatings for potential applications in biotechnological and biomedical fields. On the other hand, although there are works about the synthesis of antibacterial LbL films with AgNPs using dip-coating [19,20,21,22] and spin-coating [23,24] techniques, there is a noticeable lack of literature reporting a systematic study that directly compares both methods for the fabrication of these types of antibacterial films.

## 2. Results and Discussion

In this study, single nPSi layers produced by electrochemical anodization of silicon wafers were used as substrates for the layer-by-layer deposition of CHI-Ag and CMC. The schematic of the synthesis process is represented in Figure 1. Figure 1a shows the fabrication of nPSi substrates via photoassisted electrochemical etching, followed by a H_2_O_2_ chemical oxidation to stabilize the porous structure. The CDC and CSC methods are depicted in Figure 1b,c, respectively. In the CDC method, the obtained nPSi substrate was cyclically incubated into CHI-Ag next to the CMC solution after intermediate washes.

On the other hand, for the CSC process, CHI-Ag layers were cyclically placed LbL onto CMC by spin-coating after intermediate washes. After the required deposition cycles, samples were immersed in ASC in order to reduce Ag^+^ ions to Ag^0^ [16]. In this work, the final layers with 3.5, 6.5, 9.5, and 12.5 cycles were studied for both methods.

FESEM was used to study the morphology of the stabilized nPSi layer before the deposition of the CHI-Ag/CMC layers (Figure 2). Figure 2a shows a porous surface with an irregular shape tending to a circular form. The mean pore size estimated was found to be 17 ± 2 nm (Figure 2b). The cross-section of the stabilized nPSi layer had a thickness of ~5.3 µm (Figure 2c), where the depth of the pores is distinguishable (Figure 2d).

To gain a deeper insight into the layer deposition during a cycle, UV-Vis, FTIR (both in Figure 3), and XPS analysis (Figure 4) were performed on the first assembled layers. The complete cycle implies immersion in the ASC for the Ag reduction. UV–Vis reflectance spectra of nPSi-(CHI-Ag/CMC) composite layers fabricated by CDC and CSC are displayed in Figure 3a,b, respectively. For both kinds of deposition methods, UV-Vis spectra presented a similar behavior. Oxidized nPSi control showed almost zero reflectance in the 300–500 nm visible range and the typical interference signal of a thin film of 5 µm of thickness. However, after the first CHI-Ag polycation step (nPsi-CHI-Ag), the reflectance increased in the range of 300 nm to 500 nm, confirming that the layer assembled since the first step. Similarly, the reflectance also increased in that range after the CMC polyanion step (CMC). After the second CHI-Ag polycation step (CHI-Ag), the reflectance also increased due to the deposition of the new part of the layer. Moreover, the reflectance showed a low decrease in intensity after the chemical reduction of Ag ions by the ASC. This step is detrimental for the CHI-Ag layer, but to a limited extent, since the change of absorption does not appear to affect the underlying CMC layer. Finally, it is important to highlight that the dip identified at 320 nm is related to the absorption of CHI nanostructures [25,26]. This is why it was detected after the first CHI-Ag deposition and was also present after the rest of the steps, except for the bare nPSi control substrate.

Figure 3c,d show the FTIR spectra of the nPSi substrate and after each step of the composite formation during the first cycle of deposition for the CDC and CSC methods, respectively. The spectra of the nPSi sample displayed bands in the range of 615–666 cm^−1^, which corresponds to SiH_x_ deformation modes overlapping the silicon crystal modes [27]. At 1220 cm^−1^, a band associated with amorphous silicon dioxide (a-SiO_2_) [28] was observed, and at 1350 cm^−1^, a band attributed to the Si-O wiggling mode [29] was observed. The band at 3240 cm^−1^ corresponds to the stretching vibration mode of the OH in the silanol (Si-OH) groups [30,31]. These peaks are characteristic of a partially oxidized nPSi [27], the typical chemical oxidation of nPSi with H_2_O_2_ [32]. However, after applying the progressive polymer depositions in both techniques, these bands exhibited a decrease attributed to the electrostatic interactions between CMC and CHI-Ag. Moreover, the broadening of the band at 3240 cm^−1^ can be ascribed to the intermolecular interaction among SiO, CHI-Ag, and CMC [33]. This interaction arises from the characteristic bands of the utilized polymers falling within the range of 3200–3450 cm^−1^, associated with the O-H stretching in both polymers, as well as the N-H stretching of CHI [34]. On the other hand, the high intensity of bands in this range in the CDC spectra (Figure 3c) may be attributed to enhanced polymer penetration into the nanopores, resulting in a surface with a greater concentration of silanol groups from nPSi. Finally, in both techniques, the ASC spectra did not present any peaks due to the reduction of the ionic silver in CHI-Ag and stabilization of the obtained Ag nanoparticles [16], which absorb the IR radiation. Similar results were found when AgNPs were reduced into a CHI polymer layer; the intensities of the FTIR transmittance peaks were highly reduced [25].

The surface chemical composition of the composites was studied via XPS. Figure 4 shows the XPS spectra of nPSi-(CHI-Ag/CMC) composite layers fabricated by CDC (Figure 4a) and CSC (Figure 4b) after one complete cycle. In general, the survey spectra for both CDC and CSC methods evidenced the presence of C, O, Ag, N, and Si. The results show that the C 1s and O 1s signals increased and the Si 2p signal decreased for both the CDC and CSC methods with the increasing number of steps on the cycle due to the formation of the polymer layers on the top of the nPSi thin film. In addition, after the final step, the Ag 3d signal was more intense for the CDC than the CSC method. Figure 4c shows the XPS spectra of the Si 2p band of nPSi after the final step of the CSC and CDC methods. The two main typical contributions of the oxidized porous silicon layer can be observed on the nPSi sample: one located at 99.2 ± 0.1 eV, correlated with Si-Si bonds, and the other at 103.2 ± 0.1 eV, which correspond to the Si-O bonds [35]. After the first cycles for both methods, similar XPS spectra were obtained, suggesting that the polymer layer is formed on the top of nPSi thin film without modifying its physico-chemical properties. In order to study the formation of the Ag nanoparticles inside the polymer layer, Figure 4d shows the high-resolution XPS spectra of Ag 3d bands after the final step of the cycle for both methods. It can be observed that with the CDC technique, Ag 3d bands were more defined and had higher intensity than the CSC ones, suggesting that a higher amount of Ag was reduced by this method. However, both spectra could be fitted with the same two doublets with a spin–orbit separation of 6 eV and an area ratio of 2/3. The peaks located at 368 ± 0.2 and 374 ± 0.2 eV are elemental Ag, while the peaks shifted to lower binding energies at 367.5 ± 0.2 eV and 373.5 ± 0.2 are Ag^2+^, related to Ag_2_O [36,37,38]. These results also show that the predominant structure the oxidized silver, which is consistent with the XRD results.

To obtain thicker and more stable polymer thin films, in this work, we studied the properties of the nPSi-(CHI-Ag/CMC) composite layer formed by applying 3.5, 6.5, 9.5, and 12.5 complete cycles, respectively, using both methods, CDC and CSC. The cross-section FESEM images of nPSi-(CHI-Ag/CMC) composite layer after each cycle studied in this work by both methods are shown in Figure 5. For all the samples, the composite thin film was observed on the surface of the nPSi layer. For the CDC method (Figure 5a–d), it is noticeable that the surface layer increased its thickness after each deposition cycle, suggesting a successful LbL deposition. For 3.5 cycles (Figure 5a), the composite layer was formed almost completely by AgNPs of granular morphology with a particle size of around 100 nm. After 6.5 cycles (Figure 5b), the amount of AgNPs increased, as well as the size distribution, raising the thickness of the layer. For 9.5 cycles (Figure 5c) and 12.5 cycles (Figure 5d), the formation of a homogeneous composite polymer/AgNPs layer began to be particularly remarkable, in which the amount of AgNPs predominated on the layer. The maximum thickness of the thin film was around 0.8 µm for cycle 12.5. On the other hand, Figure 5e–h show cross-section images of the composites obtained with the CSC method at the same cycles. Similarly to the CDC method, it is evident that an increase in the number of cycles corresponds to a thicker layer, thereby confirming the success of the process. However, in contrast to the CDC method, the morphology of the AgNPs obtained by CSC was mostly in the form of flakes and very few were in the form of granules, with a wide range of sizes: from 50 to 500 nm. Furthermore, the composite layer exhibited a predominantly homogeneous distribution of the polymer, which encapsulated the AgNPs. Moreover, the maximum achieved thickness was approximately 1.3 µm at cycle 12.5, as depicted in Figure 5h.

To confirm the presence of Ag on the composite layer, XRF measurements were performed (Figure 6). The fluorescence signal was determined by measuring the intensity of the Ag Kα line (21.99 keV), which is proportional to the amount of Ag. The Ag fluorescence signal increased significantly according to the number of cycles in the samples obtained by the CDC. However, for the CSC method, the amount of Ag was clearly lower, in concordance with FESEM images. This result confirms the higher amount of AgNPS obtained by CDC compared CSC samples, as was expected from XPS analysis.

RBS spectroscopy was performed to analyze the diffusion of AgNPs in the different polymer and nPSi layers. Figure 7a,b show the RBS spectra for the composites obtained by CDC and CSC after 12.5 cycles, respectively. Four element-related signals can be observed for both samples, corresponding to C, Si, O, and Ag. The Ag signal shows a peak at the 1295 and 1306 channel numbers for CDC and CSC, respectively. These signals confirm the presence of silver in the composites, in addition to a deeper infiltration of silver in the CDC composites. On the other hand, there is a significant decrease in the intensity of the silver peaks when the composites are produced by CSC. This diminution could be attributed to a larger Ag particle size [39], or to a migration of the particles towards the film surface [40]. Other authors have associated this with a lower number of silver atoms diffusing in the composite [41].

Figure 7c,d show the depth concentration profiles of silver derived from the RBS spectra at different deposition cycles for CDC and CSC samples, respectively. The profiles obtained with the CDC method (Figure 7c) show that the Ag concentration does not vary significantly between the different layers obtained, but there is a difference between cycles 3.5 and 9.5, where the silver tends to be closer to the surface, as opposed to cycle 12.5, where it is found at a greater depth. This behavior was also observed when analyzing the CSC profiles (Figure 7d). This can be explained because the CHI/CMC polymer shielded the signal of Ag. Because both substances are weak polyelectrolytes, these systems showed exponential growth during the LbL assembly process [42].

The crystal structure of the nPSi-(CHI-Ag/CMC) composites was analyzed via X-ray diffraction. Figure 8a shows the peaks of the composites obtained with CDC. The reflection peaks at 32.95°, 38.17°, and 56.32° can be attributed to the (111) plane of Ag_2_O [43], the (111) plane of metallic silver (Ag) [43], and the (311) plane of Si, respectively. These peaks could be attributed to the fact that the silver particles obtained are in the nanometer range, which leads to a higher surface area/volume ratio. Thus, the Ag particles have high reactivity with oxygen in the air [16], leading to Ag_2_O in concordance with XPS analysis. The XRD spectra of the composites obtained with the CDC method show an increase in the intensity of the peaks when the number of cycles increases. The composites obtained by CSC (Figure 8b) show the same peaks associated with Ag, Ag_2_O, and Si, but their intensity roughly increases with the number of cycles. A similar effect was observed in the XRF characterization (Figure 6).

Figure 8c,d show the Raman spectra of nPSi-(CHI-Ag/CMC) composite layers obtained by the CDC and CSC techniques, respectively, in different deposition cycles. Figure 8c shows that the spectra of the nPSi-(CHI-Ag/CMC) composites obtained by CDC did not show significant changes in their spectra as the number of cycles increased. The main signals observed were at 2925 cm^−1^ and related to the stretching vibrations of the ν(C-H). This vibration was found in both polysaccharides (CMC and CHI) [44,45]; the band at 1644 cm^−1^ is attributed to the absorption of the carbonyl (C=O) stretching of amide I in CHI [46] at 1547 cm^−1^ bending vibrations δ(NH_2_), or at antisymmetric stretching of carboxylate group (COO^-^) and bands around 1378 cm^−1^ due to stretching vibration ν(N-H) of the amide. Figure 8d shows that the characteristic peaks of the two polysaccharides at 2922 cm^−1^, 1640 cm^−1^, 1570 cm^−1^, and 1396 cm^−1^ can be attributed to ν(C-H), ν(C=O), δ(NH_2_) and ν(N-H), respectively. The spectra decrease in intensity as the number of cycles increases; this can be attributed to the loss of the surface-enhanced Raman scattering effect (SERS), which improves the Raman scattering of a molecule when there is a metallic nanoparticle nearby [34]. In this case, the SERS effect is produced by the silver nanoparticles [9]. This effect is gradually lost, which decreases the intensity of the polymer peaks. This could be due to a lower amount of AgNPs on the surface of the composites obtained with CSC.

The nPSi-(CHI-Ag/CMC) composite layers studied in this work presented a high antibacterial rate. Tests against *E. coli* and *S. aeureus* are shown in Figure 9a and Figure 9b, respectively. In particular, both types of samples presented high antibacterial activity against the *E. coli* bacteria strain. CDC samples did not allow bacterial proliferation from 3.5 cycles, while CSC samples needed 6.5 cycles to complete the inhibition of the bacterial growth on the surface. On the other hand, CDC samples showed a high antibacterial activity against *S. aeureus* just after 9.5 cycles, whereas CSC samples merely obtained 50% of antibacterial rate after 12.5 cycles.

To obtain deeper statistical information about the antibacterial activity, Figure 9c,d present the area covered by bacteria for each sample. The area covered by *E. coli* (Figure 9c) showed a significant difference with respect to the nPSi substrate from the sample formed after 3.5 cycles. This also applied to both CDC and CSC, exhibiting a total absence of bacteria after 6.5 cycles on each sample. On the other hand, the area covered by *S. aeureus*, Figure 9d) showed only a mild difference after 3.5 cycles with respect to the control. Although the antibacterial activity increased after 6.5 cycles for both CDC and CSC, the area was totally clear of bacteria only for the CDC sample after 12.5 cycles. The superior results observed against *E. coli* can be explained because it is known that *S. aureus* has a thicker cell wall compared to *E. coli* [47]. On the other hand, although Ag flakes are used to present higher inhibition of bacteria than AgNPs because they have a higher surface-to-volume ratio [48], in our case, the antibacterial activity was independent of the shape (spherical for CDC and flakes for CSC), but the antibacterial activity of the films was found to be dependent on the amount of Ag. In this sense, CDC demonstrated superior antibacterial activity compared to CSC, attributed to the higher deposition of Ag in CDC, which was around 3–4 times higher than CSC, according to XRF and RBS results.

In a previous study conducted by our research group [16], we successfully synthesized composites of nPSi and Ag using CDC and CSC methods. However, we did not incorporate the LbL assembly of CHI and CMC in that work. As a result, the obtained composites consisted around 100 nm of AgNPs deposited on nPSi substrates, exhibiting potential applications for Surface-Enhanced Raman Scattering (SERS). In contrast, in this study, AgNPs were embedded into the polymer matrix of an LbL assembly of CHI/CMC using CDC and CSC methods, which shows that these films have potential as antibacterial surfaces.

Other authors have reported the antibacterial effect of LbL films with AgNPs obtained by dip-coating and, more recently, spin-coating. In the case of dip-coating, authors reported an excellent antibacterial effect against *E. coli* and *S. aureus*, with a slightly better effect on the *E. coli* bacteria. For example, Song et al. (2013) [20] showed the antibacterial activity of films of Ag ions and ovalbumin films LbL assembled on polyacrylonitrile nanofibrous, and Gadkari et al. (2020) [22] reported the antibacterial effect of a cotton fabric coated with polystyrene sulfonate and Ag-CHI films assembled LbL. In the case of the spin-coating technique, Li et al. (2016) [23] synthesized a hybrid coating composed of hydroxyapatite, AgNPs, and CHI on a Ti substrate; the authors also reported a higher antibacterial activity against *E. coli* than the *S. aureus* strain. Our study shows that during the synthesis of films, to increase a few the number of layers, the amount of AgNPs increased enough to reach up to 100% of the antibacterial rate of *S. aureus*. Moreover, it was also shown that this effect was achieved faster with dip-coating depositions due to the higher amount of AgNPs embedded in the films than the spin-coating technique.

## 3. Materials and Methods

### 3.1. Polyelectrolyte Solutions

Two polymers were used for the preparation of electrolyte solutions: carboxymethylcellulose (CMC, molecular weight ≈ 9 × 10^4^ g/mol, degree of substitution ≈ 0.9) and chitosan (CHI, molecular weight ≈ 5 × 10^4^ g/mol, 75–85% deacetylated) (Sigma-Aldrich, Burlington, MA, USA). Polyelectrolyte solutions were prepared by dissolving the respective polymer in distilled water at concentrations of 1% (*w*/*v*). In the case of CMC, pH was adjusted to 4. CHI solution was also prepared with 100 mM glacial acetic acid. Then, AgNO_3_ salt was dissolved in the CHI solution at a concentration of 1 mM and adjusted to pH = 4.0 (CHI-Ag). All solutions were stirred overnight.

### 3.2. Fabrication of nPSi Substrate

The fabrication of nanostructured porous silicon substrates (nPSi) was performed using the galvanostatic method by electrochemical etching of single-crystalline p-type Si wafers (boron-doped, orientation <100>, resistivity 0.001–0.005 Ω·cm) in a 1:2 electrolyte solution of hydrofluoric acid:ethanol. The applied current density was set to 80 mA/cm^2^ for 120 s. The as-prepared nPSi layers were stabilized by a chemical oxidation process in H_2_O_2_ (30% *v*/*v*) overnight, as previously reported [32]. Finally, nPSi substrates were dried under nitrogen streamflow and used as the template for silver deposition. Si wafers were purchased from University Wafer, South Boston, MA, USA, and hydrofluoric acid, ethanol, and H_2_O_2_ were acquired from Merck, Santiago, Chile.

### 3.3. Formation of the nPSi-(CHI-Ag/CMC) Composite Layers

In order to fabricate the nPSi-(CHI-Ag/CMC)_X_ composite layers (X = the number of cycles), one of two thin-film deposition techniques was used: (i) CDC or (ii) CSC. The standard deposition for the CDC series comprised the alternate immersion of nPSi substrates in the CHI-Ag and CMC solutions for 15 min; each deposition was followed by three consecutive distilled water rinse steps of 2, 1, and 1 min, respectively. The CDC assembling was repeated for 3.5, 6.5, 9.5, and 12.5 cycles in each case with the CHI-Ag coating on the top. On the other hand, the standard deposition for the CSC series consisted of alternating 500 μL of CHI-Ag and CMC solutions onto the nPSi substrate at 3000 rpm for 60 s; each deposition was followed by 500 μL of distilled water rinse step, also at 3000 rpm during 60 s. This process was repeated for 3.5, 6.5, 9.5, and 12.5 cycles. For both techniques, after samples were obtained at different cycles, they were immersed in 1 mM ascorbic acid solution (ASC) for 3 h to reduce the ionic silver and stabilize the silver nanoparticles. Finally, the samples were washed with ultrapure water at pH 4 and dried at room temperature. All chemical products were acquired from Merck, Santiago, Chile.

### 3.4. Physicochemical Characterization Techniques

The first cycle of nPSi-(CHI-Ag/CMC) via LbL depositions was monitored by UV–Vis, Fourier transform infrared (FTIR), and X-ray photoelectron spectroscopies. UV–Vis absorbance spectra were recorded using a Jasco V-750 double-beam spectrophotometer (Hachioji, Tokyo, Japan).

FTIR spectroscopy was used to identify the functional groups in the nPSi-(CHI-Ag/CMC) multilayers. The FTIR spectra were obtained using a Spectrum Two FTIR Spectrometer in the 4000–450 cm^−1^ range with 4 cm^−1^ resolution.

XPS was used to determine the surface chemical composition of the nPSi-(CHI-Ag/CMC). XPS spectra were acquired in the Surface Analysis Station 150 XPS RQ300/2 (STAIB Instruments, Munich, Bavaria, Germany) equipped with a hemispherical electron analyzer using an Mg-anode X-ray source. The pass energy was set at 20 eV, giving an overall resolution of 0.9 eV. All XPS binding energies were referenced to the adventitious C 1s carbon peak at a binding energy of 284.6 eV to compensate for the surface charging effects. The fitting of the XPS spectra was carried out using the CasaXPS 2.3.15 software.

The 3.5, 6.5, 9.5, and 12.5 cycles of nPSi-(CHI-Ag/CMC) obtained with the CDC and CSC techniques were characterized by field-emission scanning electron microscopy (FESEM), X-ray fluorescence (XRF), Rutherford backscattering spectroscopy (RBS), X-ray diffractometry, and Raman spectroscopy.

The surface and cross-sectional morphologies of the samples were observed by FESEM (Philips XL-40FEG, Cambridge, MA, USA), operating with an acceleration potential of 10 keV. Images were obtained from the FESEM images that were processed using freely available ImageJ 1.52k software.

XRF measurements were carried out using an irradiation system consisting of an X-ray generator (Teledyne ICM model 160D) with the ability to operate at voltages of 10–160 kV, tube currents of 1.0–10 mA, and 900 W of power. The generator had a Tungsten (W) anode and 800 μm Beryllium window. The equipment was configured for irradiations with voltages of 50 kV and 2.0 mA. A 0.8 cm long lead collimator and an internal hole of 3 mm in diameter permitted the irradiation of a circular area of 1.5 cm diameter of the analyzed samples at a tube-sample distance of 15 cm. The detection system consisted of a Canberra low-energy germanium detector (LE-Ge) model GL1010 based on semiconductor diodes with a P-I-N structure, with the intrinsic region (I) sensitive to ionizing radiation, particularly X-rays and γ-rays from 3 keV to 2 MeV. The GL1010 had a detector crystal with an area of 1000 mm^2^ and 1 cm thickness and was characterized by an energy resolution of about 300 eV at 5.9 keV. The LE-Ge detector was operated with the digital pulse processor (DPP) AMPTEK^®^ model HPGe-PX5.

In-depth profiling of the AgNPs infiltration into nPSi layers was studied using RBS. RBS experiments were carried out at the standard beamline of the Centro de Micro-Análisis de Materiales (CMAM), which hosts a 5 MV Cockroft–Walton tandetron accelerator. For the measurements, 2 MeV He^+^ ions were used. The scattered ions were detected at a scattering angle of 170° with a Si semiconductor particle detector; the samples were oriented in random geometry to avoid channeling through the crystalline substrate. The vacuum was about 5 × 10^–5^ Pa. Simulations and spectra fitting were carried out using SIMNRA 7.02 software.

The crystalline structures of nPSi-(CHI-Ag/CMC) composite layers were examined using a Smartlab X-ray diffractometer, with Theta-2 Theta Bragg-Brentano geometry and solid-state D/teX Ultra 250 detector (Rigaku Corporation, Tokyo, Japan). The instrumental alignment was checked against the NIST SMR 660c LaB6 powder standard, and its optic configuration employed Ni-filtered Cu radiation (30 kV and 40 mA), 0.5° divergence slit, 0.25° anti-scatter slit, and both sides with 5° Soller slits. In preference, patterns were collected in the 10–60° range, counting 0.5°/s per step of 0.01°. PDXL 2 v.2.7.3.0 software and ICDD 2018 PDF-4 reference database were used for matching.

The Raman spectra were acquired at room temperature using a Renishaw Ramascope 2000 microspectrometer, in the range between 0 and 4000 cm^−1^, and a 514.5 nm excitation wavelength (green) line from an argon-ion laser. Exciting light was focused on the sample surface with a BH-2 Olympus microscope. The objective had a 50× magnification and a numerical aperture of NA = 0.85. The laser power on the sample surface was in the order of 1 mW. The integration time for each CCD pixel was 50 s.

### 3.5. Antibacterial Assays

The antibacterial activity of nPSi-(CHI-Ag/CMC)_x_ composite layers was studied against *Escherichia coli* (Gram-negative) and *Staphylococcus aureus* (Gram-positive) with the well diffusion method using Muller–Hilton agar. Both bacteria strains were grown overnight at 37 °C with a standard turbidity concentration of McFarland 0.5 (10^8^ CFL/mL). Pieces of 5 mm × 5 mm were incubated in triplicate in the well plates for 24 h at 37 °C without culture media replacement. Micrographs of the surface of composite layers after 24 h of contact with both kinds of bacteria strains were obtained with a CMEX-10 Pro Microscope camera coupled with a JNOEC JSZ4 binocular loupe, and the area covered by bacteria was estimated by analyzing the images with ImageJ 1.52k software. nPSi layers were used as control.

The antibacterial rate was calculated as:(1)R(%)=((A−B)/A)·100
where *A* is the covered area by bacteria on the control sample (nPSi substrate), and *B* is the mean fraction area of bacteria adhered to the composite’s layers.

## 4. Conclusions

In this work, films of AgNPs embedded into CMC/CHI multilayers assembled layer-by-layer on nPSi were fabricated with two types of deposition techniques: cyclic dip-coating (CDC) and cyclic spin-coating (CSC). The first cycle of deposition was monitored by UV–Vis, FTIR, and XPS. The results showed AgO and Ag formation from the first obtained layers using both techniques. After 12.5 cycles, each process produced a specific thickness of film: 0.8 µm for CDC and 1.3 µm for CSC. Moreover, the embedded AgNPs also showed differences: CDC generated granular particles with a size of 100 nm, and CSC produced flakes and granules within a size range of 50–500 nm. On the other hand, XRF exhibited the relative Ag concentration on the samples; meanwhile, RBS showed the Ag in-depth distribution inside the polymeric matrix. XRD diffractograms evidenced no crystallographic differences of the Ag nanoparticles incorporated by both deposition techniques, as the Ag_2_O phase was predominant over Ag. Samples were also chemically characterized by Raman spectroscopy, and the results showed a constant composition of films during the deposition cycles. In general, the CDC technique showed a higher deposition of AgNPs than CSC, which favored an elevated antibacterial activity. From the third cycle, the CDC method was able to inhibit *E. coli* and *S. aureus* bacteria adhesion. These results show that these films with AgNPs embedded into CMC/CHI multilayers on nPSi have the potential for biomedical purposes as an advantageous antibacterial coating on structural biomaterials.

## Figures and Tables

**Figure 1 ijms-24-10595-f001:**
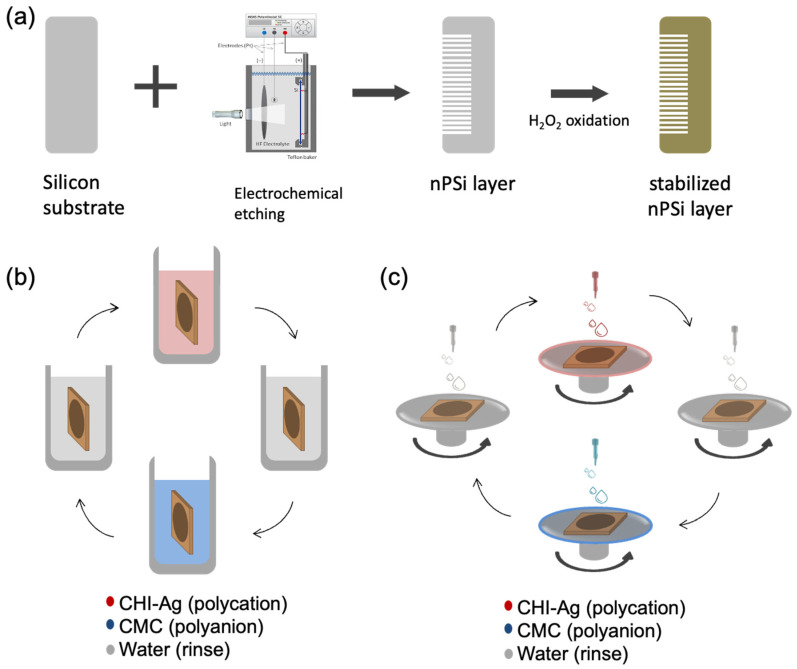
Schematic illustration of the nPSi-(CHI-Ag/CMC) composite layers: (**a**) synthesis of nPSi substrate via electrochemical etching and stabilization by chemical oxidation, (**b**) CHI-Ag and CMC deposition onto nPSi by cyclic dip-coating (CDC), and (**c**) CHI-Ag and CMC deposition onto nPSi by cyclic spin-coating (CSC). In both cases, after the required deposition cycles were performed, samples were immersed into an ascorbic acid solution (ASC) for Ag reduction.

**Figure 2 ijms-24-10595-f002:**
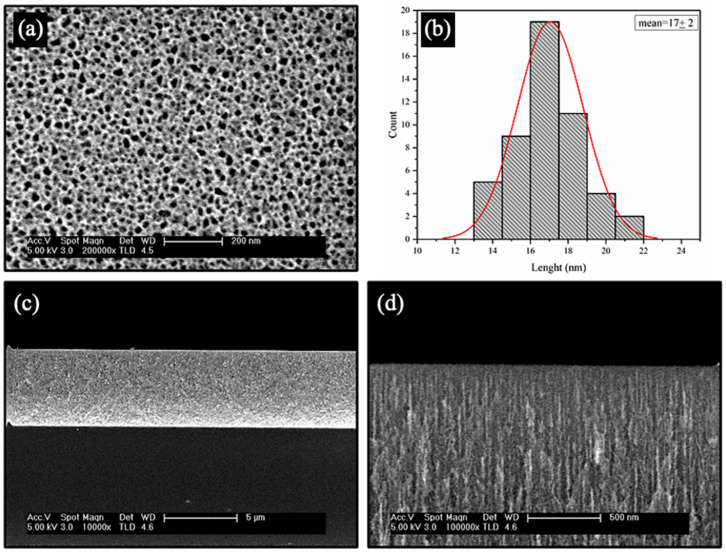
FESEM images of a nPSi layer used in this study: (**a**) surface and (**b**) histograms of particle size distribution, (**c**,**d**) cross-sectional view.

**Figure 3 ijms-24-10595-f003:**
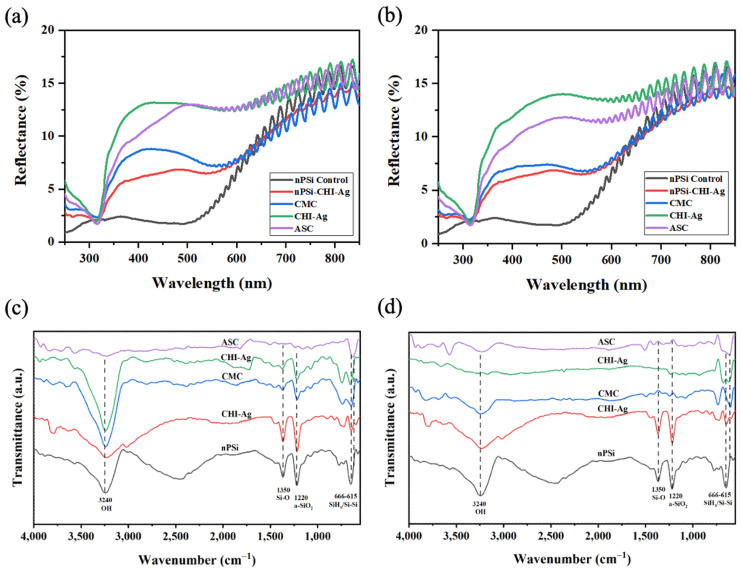
Reflectance spectra of nPSi-(CHI-Ag/CMC) composite layers fabricated by (**a**) CDC and (**b**) CSC. FTIR-ATR spectra of nPSi-(CHI-Ag/CMC) composite layer fabricated by (**c**) CDC and (**d**) CSC.

**Figure 4 ijms-24-10595-f004:**
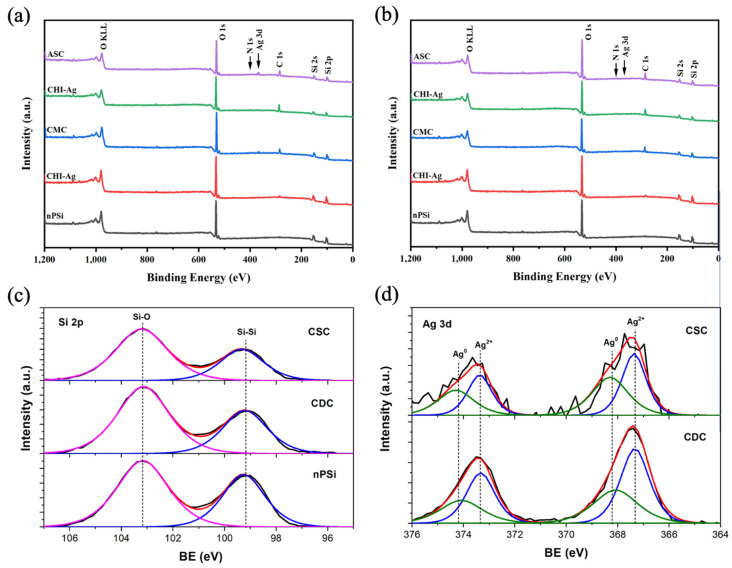
XPS survey spectra of nPSi-(CHI-Ag/CMC) composite layers fabricated by: (**a**) CDC and (**b**) CSC. (**c**) XPS spectra of Si 2p bands of nPSi and after ASC step for CSC and CDC methods. (**d**) XPS spectra of Ag 3d bands of CSC and CDC layer after ASC step.

**Figure 5 ijms-24-10595-f005:**
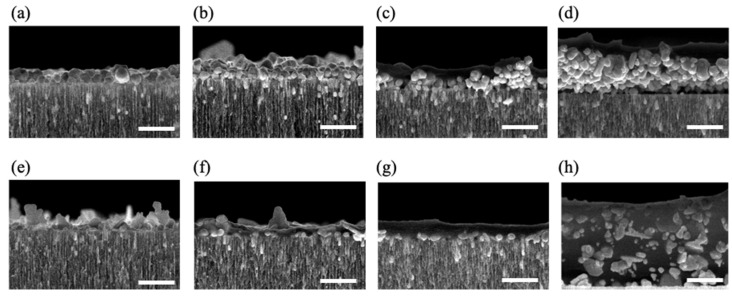
Cross-sectional FESEM images of a typical nPSi-(CHI-Ag/CMC) composite layer after deposition using CDC: (**a**) 3.5 cycles, (**b**) 6.5 cycles, (**c**) 9.5 cycles, and (**d**) 12.5 cycles; and CSC: (**e**) 3.5 cycles, (**f**) 6.5 cycles, (**g**) 9.5 cycles, and (**h**) 12.5 cycles. Scale bar: 500 nm.

**Figure 6 ijms-24-10595-f006:**
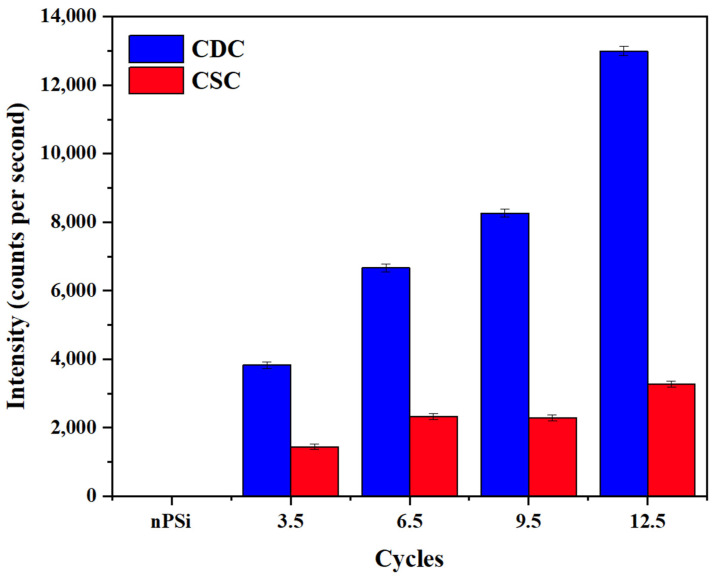
Ag fluorescence signal in nPSi-(CHI-Ag/CMC) composite layers after 3.5, 6.5, 9.5, and 12.5 CHI-Ag and CMC deposition cycles using CDC and CSC techniques.

**Figure 7 ijms-24-10595-f007:**
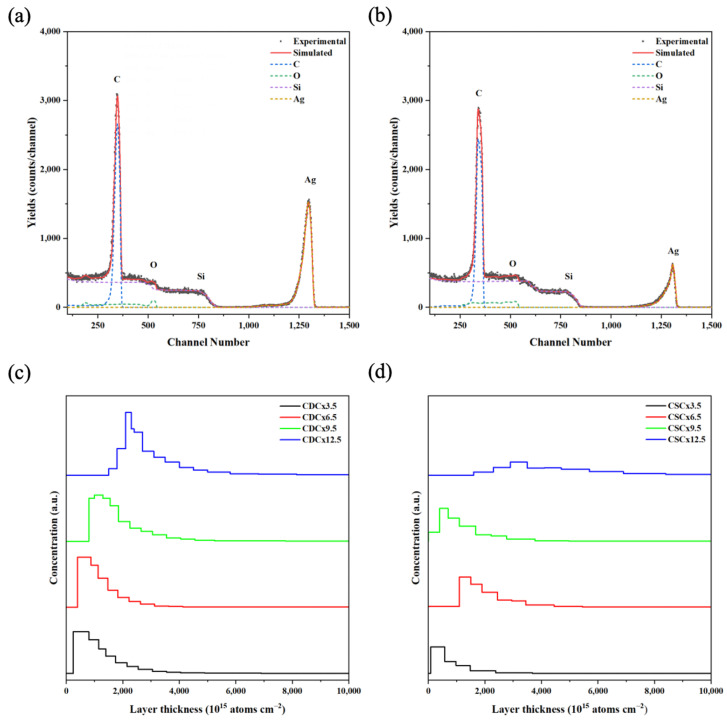
RBS spectrum and in-depth concentration profile of Ag obtained via simulation of typical nPSi-(CHI-Ag/CMC) composite multilayers by (**a**,**c**) CDC, and (**b**,**d**) CSC.

**Figure 8 ijms-24-10595-f008:**
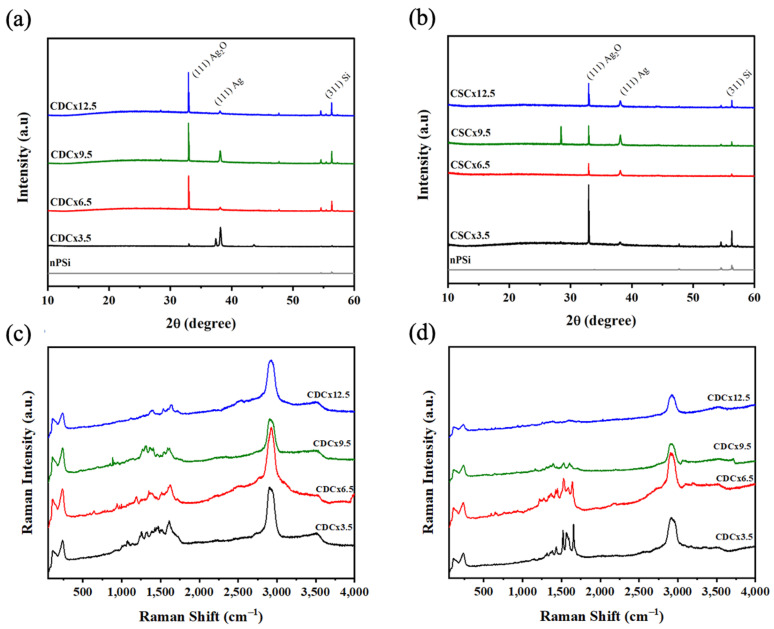
XRD pattern of nPSi-(CHI-Ag/CMC) composite layers after 3.5, 6.5, 9.5, and 12.5 CHI-Ag and CMC deposition cycles using: (**a**) CDC and (**b**) CSC. Raman spectra for the nPSi-(CHI-Ag/CMC) composite layer after 3.5, 6.5, 9.5, and 12.5 CHI-Ag and CMC deposition cycles using: (**a**) CDC and (**b**) CSC.

**Figure 9 ijms-24-10595-f009:**
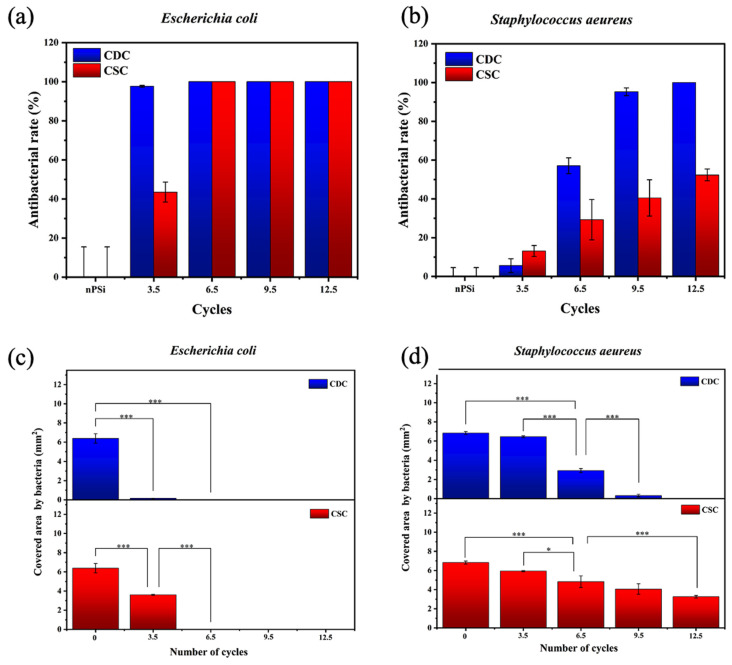
Antibacterial rate of nPSi-(CHI-Ag/CMC) composite layers after 3.5, 6.5, 9.5, and 12.5 CHI-Ag and CMC deposition cycles using CDC and CSC techniques against (**a**) *E. coli* and (**b**) *S. aeureus*. nPSi layers were used as control. Covered area by (**c**) *E. coli* and (**d**) *S. aeureus* on nPSi-(CHI-Ag/CMC) composite layers after 0, 3.5, 6.5, 9.5, and 12.5 cycles for CDC and CSC techniques. Error bars represent mean ± S.D. of three measurements, and they were statistically interpreted by analysis of variance (ANOVA) multifactorial model, where * and *** denote a significant difference compared to the nPSi substrate after 24 h with *p* < 0.05 and *p* < 0.001, respectively.

## Data Availability

The data that support the findings of this study are available from the corresponding author upon reasonable request.

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
