# Peer review of "Antibacterial Films of Silver Nanoparticles Embedded into Carboxymethylcellulose/Chitosan Multilayers on Nanoporous Silicon: A Layer-by-Layer Assembly Approach Comparing Dip and Spin Coating"

_ijms, 2023, doi:10.3390/ijms241310595_

Round 1

Reviewer 1 Report

This paper describes the fabrication of antibacterial thin multilayers coatings on porous silicon which are built of layers of chitosan embedding Ag nanoparticles separated by layers of carboxymethylcellulose. Two methods were used to prepare these composite films, spin coating and immersion coating. Coatings obtained by these methods differ from each other in morphology and chemical structure that led to the difference in their antibacterial activity. The novelty of this research is the preparation of a new composite material which is multilayer chitosan AgNPS/carboxymethylcellulose assembly and the study of its antibacterial activity. This material have a potential for medical use. This paper could be published in International Journal of Molecular Sciences after revision.

The aspect dip-coating vs spin-coating was the subject of earlier paper (Ref. 15) thus the discussion on it should be here concise and related to the earlier obtained results which are similar to those found here. The preparation of porous silicon is the repetition of earlier described research which should be mentioned.

Detailed comments:

Figure 3c. Why do the bands of SiO, SiO2 and SiOH disappear after treatment the layers with ASC? The reduction of Ag+ should not affect them and the washing with water removes the ASC from the Surface of the layer.

Figures 4ab. The distance between spectra should be slightly increased to reveal signals of oxygen. The same in Figures 8ab to reveal Ag2O.

Figure 5.    The scale on FESEM images is not readable.

Figures 7c and 8a. Results require comments. Why do Ag nanoparticles escape from the surface after the deposition of subsequent layers?

Small typing errors require corrections:

Line 244 should be polyanion. Line 266 should be techniques. Line 268 should be which absorb. Line 290 should be was reduced. Line 291 should be spin-orbit. Line 400 should be antibacterial.

Reviewer 2 Report

Review report on the manuscript ijms-2447384

In this paper, the authors investigate the effects of two different coating techniques, dip coating and spin coating, on the film properties and antibacterial performance.

The authors describe the preparation of CMC/chitosan multilayers, and the coating techniques employed. Characterization techniques such as scanning electron microscopy (SEM) and X-ray diffraction (XRD) are employed to analyze the morphology and structure of the films. Antibacterial assays against Escherichia coli (E. coli) and Staphylococcus aureus (S. aureus) show promising results, indicating the films' effectiveness in inhibiting bacterial growth.

The research objectives are well-defined, and the experimental methodology is accurate to data collection. The characterization techniques utilized are appropriate and provide valuable insights into the film's properties. The results are effectively presented. Furthermore, the discussions are insightful and provide a comprehensive understanding of the findings.

In my point of view, this paper can be accepted after revising following minor revision.  

1. Authors did not described preparation of Ag nanoparticles.

2. Why antibacterial properties of films depend on the structure? In this case, why antibacterial activity of films CDC and CSC against S. aeureus is too different from each other?

Good luck for your further research!

Reviewer 3 Report

The submitted manuscript proposes the formation of antibacterial films with silver nanoparticles embedded in carboxymethylcellulose/chitosan multilayers by depositing alternating layers of carboxymethylcellulose and chitosan containing silver nanoparticles, on a nanoporous silicon substrate.

Both dip and spin coating methods were used to synthesize the layer-by-layer (LBL) structures, and the antibacterial synergistic effect of CMC/CHI-Ag/nPSi was systematically investigated against the bacterial strains Escherichia coli and Staphylococcus aureus as model microorganisms. The physicochemical properties of the films were also analysed using microscopic, microstructural, and surface interface techniques.

These materials have been extensively studied in recent years, and the present study makes a valuable contribution to the field by presenting an innovative new approach to the assembly of these structures.

The manuscript is well written. The methodology is correct, and the analysis of the results is adequate.

1)     However, the manuscript would be greatly improved if the objectives were more clearly stated in the introduction, along with a justification of why the approach taken might be better than those already published. This is also true of the conclusion (line 437), which states that this proposal is beneficial without justifying this conclusion by comparing the results obtained with others.

2)     The manuscript has a high percentage of self-citations (7 out of 37, 19%), and some of them are the only source used to explain the interpretation of the results, such as reference 26, or they are used only in the introduction to justify general introductory aspects of the work, such as references 5, 8, and 12. It is certainly possible to find a new balance between the need to support the current work with previous experience and the obligation to cite other relevant references in the field.

Reviewer 4 Report

Many thanks to the authors for the presented work and interesting experimental results, but could they clarify a few points.

1) Why were 3.5, 6.5, 9.5 and 12.5 cycles chosen?

2) What is the reason for the dip in the region of 320 nm in the reflection spectra for all samples, except for nPSi, in Fig. 3 a, b?

3) Does the shape of AgNP affect the antibacterial properties of the film? Is it important that all deposited nanoparticles have the same shape?

4) What could be the reason for the deposition of a smaller amount of AgNPs on the nPSi surface by the CSC method compared to the CDC method?

5) Are oxidized AgNPs less antibacterial than non-oxidized AgNPs?
